# Advanced Applications for Protein and Compounds from Microalgae

**DOI:** 10.3390/plants10081686

**Published:** 2021-08-16

**Authors:** Daniela Castiglia, Simone Landi, Sergio Esposito

**Affiliations:** 1Bio-Organic Chemistry Unit, Institute of Biomolecular Chemistry CNR, Via Campi Flegrei 34, 80078 Pozzuoli, Italy; d.castiglia@icb.cnr.it; 2Department of Biology, University of Naples “Federico II”, Via Cinthia, 80126 Napoli, Italy; sergio.esposito@unina.it

**Keywords:** microalgae, diatoms, biodiesel, synthetic biology, bioreactor, extreme environments

## Abstract

Algal species still show unrevealed and unexplored potentiality for the identification of new compounds. Photosynthetic organisms represent a valuable resource to exploit and sustain the urgent need of sustainable and green technologies. Particularly, unconventional organisms from extreme environments could hide properties to be employed in a wide range of biotechnology applications, due to their peculiar alleles, proteins, and molecules. In this review we report a detailed dissection about the latest and advanced applications of protein derived from algae. Furthermore, the innovative use of modified algae as bio-reactors to generate proteins or bioactive compounds was discussed. The latest progress about pharmaceutical applications, including the possibility to obtain drugs to counteract virus (as SARS-CoV-2) were also examined. The last paragraph will survey recent cases of the utilization of extremophiles as bio-factories for specific protein and molecule production.

## 1. Introduction

In recent years, public opinion, research community, and commercial stakeholders paid great attention to green and sustainable biotechnologies; these are not anymore considered a luxury, but a necessity in all countries. Actually, the current world population is estimated in 7.8 billion people, and an increase is expected of about 2 billions in the next 30 years (UN; www.un.org; accessed on 1 July 2021); therefore, it is essential to develop sustainable biotechnologies and strategies to face the increased food demand with a reduced impact on natural environment. Plants and algae possess still untouched possibilities; thus, these organisms are emerging as formidable sustainable tools with a number of advantages over the conventional biotechnologies [1,2,3,4].

Since 1980s, plants emerged as one of the most promising production platforms for recombinant proteins and for the valuable production of bio-compounds [5,6]. The agricultural scale cultivation represents a cost-effective method to produce recombinant proteins as well as an innovative approach to reduce risks of product contamination [2]. Encouraging results in the production of recombinant proteins and molecules for pharmaceutical applications were preliminarily obtained using higher plants [7]. On the other hand, the intensive exploitation of cropland represents a severe constraint influencing the environment utilization, CO_2_ emission, chemical pollution, and water quality and availability [8].

At the same time, microalgae overcome a number of limitations of plants, emerging as effective biotechnological platforms and giving added value by the exploitation of bioactive compounds obtained from these biomasses [9,10]. Microalgae are characterized for being unicellular and their versatile metabolism, representing efficient and economic platforms to gather organics compounds such as proteins, lipids, pigments, sterols, and carbohydrates for a number of commercial applications such as nutraceutical, pharmaceutical and for biofuels [11,12]. This group includes photosynthetic prokaryotes (e.g., cyanobacteria) or eukaryotes, which are able to live in diversified environments [9]. As consequence, a number of microalgae species are currently cultivated for commercial and industrial aims, and more than 75% of this production is related to healthier supplements for human consumption [10].

Furthermore, these organisms can be utilized for advanced synthetic biology applications. This represents a new experimental field, connecting molecular biology with bioengineering [13]. Recent papers reported the ability of microalgae for the management of metabolism by synthetic biology approaches [13,14]. These studies further strengthen the idea of the utilization of microalgae as versatile systems for sustainable and effective biotechnological applications.

In this review, we analyze some recent examples about the utilization of photosynthetic organisms from marine environments, as bio-factories. Furthermore, we reported a survey of the latest and advanced applications of proteins and bioactive compounds from microalgae.

## 2. Microalgae Engineering to Obtain Platforms for Biocompounds Productions

In recent years macroalgae and microalgae highlighted a renewed commercial interest about the production of natural metabolites for nutraceutical, pharmaceutical, energetic, and food industries purposes [11,12,15,16,17].

Microalgae are considered promising and sustainable platforms to produce recombinant proteins; these aquatic organisms show several advantages with respect to higher plants, such as safety, low-cost production and the ability to be grown rapidly using freshwater, seawater and wastewaters, thus avoiding arable land [18].

Microalgae cultivations represent an alternative and renewable source to produce biofuels [19,20]. In fact, microalgae produce high levels of lipids, carbohydrates and proteins, which make them a feasible candidate for ecofriendly third-generation biofuels production [11,12]. Unfortunately, the actual costs of microalgal biofuels are still considered higher than conventional fossil oils, therefore, an experimental progress is still necessary to improve and modulate the bio-compounds content. In this context, genetic engineering represents an effective approach in order to obtain a better performance from microalgae [21] (Figure 1).

Thus, a number of strategies have been used to hyperaccumulate fatty acids and/or triacylglycerol (TAG) content (Table 1). *Phaeodactylum tricornutum* and *Thalassiosira pseudonana* engineered strains were obtained modifying TAG biosynthetic pathways. In this context, critical enzymes are glycerol-3-phosphate acyltransferase (GPAT), lysophosphatidic acid acyl- transferase (LPAAT), and acyl-CoA:diacylglycerol acyltransferase (DGAT). These enzymes regulate fatty acid and TAG biosynthetic pathways in plants and microalgae [22,23,24]. The overexpression of GPAT is effective to obtain suitable strains for biofuel production (Table 1). Both isoforms GPAT1 and GPAT2 were overexpressed in *P. tricornutum* resulting in a 2.9- and 2.3-fold change increased lipid content, respectively [19,20]. Interestingly, the overexpression of GPAT2 regulated the natural expression of LPAT and DGAT, causing an enhanced effect on total lipid content [20]. Interestingly, this strain was also reported as better tolerant to abiotic stress, namely salinity and chilling [20]. A similar approach was recently used in *Neochloris oleoabundans* where the plastidial *Neo*LPAAT1 and endoplasmic reticulum-located *Neo*DGAT2 were co-overexpressed. This strategy increased the TAG content of about 2-fold compared to wild type; furthermore, engineered strain reporting no effects on algal biomass [25]. Significant results were obtained using heterologous expression, mutagenesis in situ and knockdown strategies of genes related to the lipid biosynthetic pathway. Trentecoste et al. [26] and Barka et al. [27], developed specific lipase knockout strains of *T. pseudonana* and *P. tricornutum,* respectively. The suppression of TAG lipase in these two diatoms significantly increased the lipid content. Interestingly both did not show effects on growth [12,26,27]. TALEN-based genome editing technique was used in *P. tricornutum* strain [28] reporting modification in *Pt*TES1, a hotdog-fold thioesterase involved in acyl-CoA hydrolysis. This modification lead to a 1.7-fold change increase of TAG content [28]. Successful editing results were obtained by using CRISP/CAS9 approach on *P. tricornutum* and *T. pseudonana*. These preliminary results opened new perspectives to obtain improved microalgae strains using this technique [29,30,31]. 

An increase in lipids content has been obtained by modifying different microalgal pathways. LEAFY COTYLEDON1 from *Arabidopsis thalian*
*(At*LEC1) represents a key transcription factor involved in plants lipid metabolism [32]. Liu et al. [33] demonstrated that the endogenous expression of *At*LEC1 improves fatty acids and total lipids content in *Chlorella ellipsoidea* both under mixotrophic and autotrophic culture conditions. These differences between wild type and the engineered algal strain were related to an enhanced regulation of key enzymes namely ACCase, GPDH, PDAT1, and DGAT1 [33]. Another interesting example of the transfer of genes from *A. thaliana* in microalgae showed the heterologous overexpression of 1-deoxy-D-xylulose 5-phosphate synthase from *A. thaliana* in *Nannochloropsis oceanica*. This strain reported an improved CO_2_ fixation, thus increasing biomass, lipids, and carbohydrates productions upon different nutritional and stressed conditions [34]. Particularly, *Nannochloropsis* represented a further model used as platform for biofuel production from microalgae [34,35,36,37]. In the last year a number of papers have been published about the genetic manipulation of *Nannochloropsis*, particularly focused on the increase on lipid and fatty acids content. Different approaches, namely overexpression [36], endogenous expression [37], and insertional mutagenesis [38] were successfully used in *Nannochloropsis*. For example, Jeon et al. [36] overexpressed the NADP-dependent malic enzyme (*Ns*ME1) in *Nannochloropsis salina.* This enzyme is an important NADPH supplier, playing the central role in the C4-like cycle of microalgae. The overexpressed strain showed an increased fatty acid and lipids contents compared with wild type, up to 53% and 38%, respectively [36].

Recently, several papers reported the role of glucose-6-phosphate dehydrogenase (G6PDH) in lipogenesis regulation [38,39]. This enzyme catalyzes the first and rate-limiting reaction of the oxidative pentose pathway (OPPP) playing critical role(s) in both plants and algae physiology [40], particularly upon nutrient starvation and abiotic stresses [38,41]. G6PDH was efficiently used for homologous overexpression and heterologous expression in microalgae. The increased NAPDH content in *P. tricornutum* induced by *Pt*G6PDH overexpression stimulated an increase fatty acid production upon control and high CO_2_ level conditions [39]. Similarly, heterologous expression of *Pt*G6PDH and *No*G6PDH (from *Nannochloropsis oceanica*) in *Chlorella pyrenoidosa* showed an increase in lipid content of about 1.79 and 3.09-fold-change, respectively.

Among lipids, long-chain polyunsaturated fatty acid (PUFA, ω-3) represent a major fraction of microalgae biomass such as *Cyclotella cryptica*, *Nitzschia laevis*, *Navicula incerta*, *Navicula pelliculosa*, and others [42,43,44]. Particularly, eicosapentaenoic acid (EPA 20:5) and docosahexaenoic acid (DHA 22:6) showed an interesting commercial potential [42,43,44]. Homologous overexpression and heterologous expression approaches have been investigated for an enhanced production of ω-3 species (Table 1). It is worth remembering that *P. tricornutum* represents a model organism for diatoms, mostly studied for biomass, lipid, and EPA production [43]. Recently, *P. tricornutum* overexpressing malonyl CoA-acyl carrier protein transacylase (*Pt*MCAT) and fatty acid desaturase 5b (*Pt*FAD5b) significantly improved the content of EPA [45]. Similarly, Hamilton et al. [46], obtained an engineered *P.tricornutum* strain expressing Δ-5 elongase from *Ostreococcus tauri*, and a glucose transporter from the moss *Physcomitrella patens*. This strain accumulated ω-3 long-chain polyunsaturated fatty acid and showed an unconventional ability to grow in the dark [46]. Recently, aquaculture industry has started to replace fish oil and fish meal with plant seed meals and vegetable oils but the critical constraints in the use of this type of oils is the presence of anti-nutritional factors namely phytate [47]. Pudney et al. [48], proposed the heterologous expression of phytases from *Aspergillus niger* and *Escherichia coli* to overcome this constraint. *P. tricornutum* strains expressing the phytases from these two species showed an enhanced content of EPA and DHA [48]. 

Successful manipulations were also obtained using red microalgae as well as *Porphyridium* sp. [49,50]. These are important marine single-cell economic microalgae suitable for large-scale production of phycoerythrin, polysaccharides and PUFAs [49]. There are few cases of *Porphyridium* sp. engineered strains, mainly obtained by *Agrobacterium*-mediated and biolistic transformation methods [46]. Interestingly, Jeon et al. [50], recently reported the use of CRISP/CAS9 approach to manipulate *Porphyridium* sp. increasing the production of phycoerythrin. The authors manipulated the chlorophyll synthase gene (CHS1), obtaining an improved strain able to produce an increased content of phycoerythrin from 44% to 63% depending by the light sources [50]. 

Analogous to phycoerythrin, astaxanthin is a desirable product for human consumption. This is mainly produced by the green microalga *Haematococcus pluvialis* in response to adverse conditions namely excess of light, salinity, and nutrient starvation [51]. A number of mutagenesis approaches were used on *H. pluvialis* using physical and chemical mutagens such as UV radiation, ethyl methanesulphonate (EMS), diethyl sulphate (DES) and other. In some case desirable improved strains were obtained showing high growth rates at the vegetative stage and high astaxanthin accumulation rates at the encystment stage [51]. In recent years, genetic engineering approaches were also obtained in *H. pluvialis*. For example, Waissman-Levy et al. [52], manipulated the nuclear genome of *H. pluvialis* by insertion of the hexose uptake protein (HUP1) gene from the green microalga *Parachlorella kesslerii*. The engineered strain was able to grow upon heterotrophy conditions in glucose-supplemented media [52].

Apart from the utilization of microalgae for the production of biomass, biofuels (e.g., lipids), and recombinant proteins, the exploitation of these organisms as makers of proteins and complex molecules of pharmacological interest is a major theme of current research. The flexible metabolism of photosynthetic cells represents an intriguing tool for the synthesis of those compounds requiring complex biosynthesis not always possible in a single cell compartment as in bacteria.

**Table 1 plants-10-01686-t001:** List of microalgae engineered strains

Host Organisms	Gene	Donor Organism	Enzyme	Approach	Effects	References
*Phaeodactylum tricornutum*	*Pt*GPAT2	*P. tricornutum*	Glycerol-3-phosphate acyltransferase 2	Overexpression	Hyperaccumulation of TAG	[20]
*Phaeodactylum tricornutum*	*Pt*PGM	*P. tricornutum*	Phosphoglucomutase	Overexpression	Increased synthesis of chrysolaminarin	[12]
*Phaeodactylum tricornutum*	*Os*Elo5	*O. tauri*	Δ5-elongase	Endogenous expression	Improved accumulation of EPA and DHA—dark cultivation	[46]
*Phaeodactylum tricornutum*	*Pp*GT	*P. patens*	Glucose transporter	Endogenous expression	Improved accumulation of EPA and DHA—dark cultivation	[46]
*Phaeodactylum tricornutum*	*Pt*TL	*_*	TAG lipase	Knockdown	Hyperaccumulation of TAG	[27]
*Thalassiosira pseudonana*	*Tp*TL	*_*	TAG lipase	Knockdown	Hyperaccumulation of TAG	[26]
*Phaeodactylum tricornutum*	*Pt*DGAT2B	*P. tricornutum*	2 acyl-CoA:diacylglycerol acyltransferase	Endogenous overexpression	Increased DHA and TAG content	[53]
*Phaeodactylum tricornutum*	*Pt*G6PDH	*P. tricornutum*	Glucose-6-phosphate dehydrogenase	Overexpression	Enhanced lipid and w-3 accumulation	[39]
*Phaeodactylum tricornutum*	*An*PhyA	*A. niger*	Phytase	Endogenous expression	Improved accumulation of EPA and DHA	[48]
*Phaeodactylum tricornutum*	*Ec*AppA	*E. coli*	Phytase	Endogenous expression	Improved accumulation of EPA and DHA	[48]
*Phaeodactylum tricornutum*	*Pt*MCAT	*P. tricornutum*	Malonyl CoA-acyl carrier protein transacylase	Overexpression	Improved accumulation of EPA	[45]
*Phaeodactylum tricornutum*	*Pt*FAD5b	*P. tricornutum*	Fatty acid desaturase 5b	Overexpression	Improved accumulation of EPA	[45]
*Phaeodactylum tricornutum*	*Pt*GPAT1	*P. tricornutum*	Glycerol-3-phosphate acyltransferase	Overexpression	Increased lipid content	[19]
*Phaeodactylum tricornutum*	*Pt*LPAAT1	*P. tricornutum*	Lysophosphatidic acid acyltransferase	Overexpression	Increased lipid content	[19]
*Phaeodactylum tricornutum*	*Pt*PTP	*P. tricornutum*	Plastidial pyruvate transporter	Overexpression	Increased production of biomass and lipids	[54]
*Phaeodactylum tricornutum*	*Pt*TES1	*_*	Hotdog-fold thioesterase	TALEN- mutagenesis	Hyperaccumulation of TAG	[28]
*Chlorella ellipsoidea*	*At*LEC1	*A. thaliana*	Leafy cotyledon 1 transcription factor	Endogenous expression	Lipid overexpression	[33]
*Synechocystis* sp.	*Sa*ACC	*S. alba*	Acetyl-CoA carboxylase	Endogenous expression	Lipid overexpression	[55]
*Chlamydomonas reinhardtii*	*Cr*GAPDH	*Ch. reinhardtii*	Glyceraldehyde-3-phosphate dehydrogenase	Overexpression	Enhanced carbon fixation	[56]
*Chlorella vulgaris*	*Cv*NR	*_*	Nitrate reductase	CRISP-cas9 editing	Reduced growth upon specific conditions	[57]
*Chlorella vulgaris*	*Cv*APT	*_*	Adenine phosphoribosyltransferase	CRISP-cas9 editing	Reduced growth upon specific conditions	[57]
*Nannochloropsis oceanica*	*At*DXS	*A. thaliana*	1-deoxy-D-xylulose 5-phosphate synthase	Endogenous expression	Improved CO_2_ absorption, biomass and lipids	[34]
*Nannochloropsis salina*	NsME	*N. salina*	Malic enzyme	Overexpression	Increased production of lipids and fatty acids	[35]
*Nannochloropsis salina*	CrLCIA	*Ch. reinhardtii*	Anion transporter	Endogenous expression	Increased production of fatty acids	[36]
*Nannochloropsis oceanica*	NoAPL	-	Apetala 2 like transcription factor	Insertional mutagenesis	Increased production of lipids	[37]
*Chlamydomonas reinhardtii*	*Cr*SBP1	*C. reinhardtii*	Sedoheptulose-1,7-bisphosphatase	Overexpression	Photosynthetic and growth rates improvement	[58]
*Neochloris oleoabundans*	*No*GPAT	*_*	Glycerol-3-phosphate acyltransferase	Overexpression	Increased lipid content	[25]
*Neochloris oleoabundans*	*No*LPAAT	*_*	Lysophosphatidic acid acyltransferase	Overexpression	Increased lipid content	[25]
*Porphyridium purpureum*	*PpCHS1*	*_*	Chlorophyll synthase	CRISP-cas9 editing	Increased phycoerythrin content	[50]
*Haematococcus pluvialis*	*PkHUP1*	*Parachlorella kesslerii*	Hexose uptake protein	Endogenous overexpression	Dark cultivation	[52]
*Chlorella pyrenoidosa*	*Pt*G6PDH	*P. tricornutum*	Glucose-6-phosphate dehydrogenase	Endogenous overexpression	Increased lipid content	[38]
*Chlorella pyrenoidosa*	*No*G6PDH	*N. oceanica*	Glucose-6-phosphate dehydrogenase	Endogenous overexpression	Increased lipid content	[38]

## 3. Microalgae Engineering for the Production of Pharmacological Proteins

Microalgae represent an important biotechnology resource for the production of recombinant proteins for pharmacological applications. Successful results were reported by the genetic manipulation of microalgae for the production of antigens for vaccines, antibodies, immunotoxins, hormones, and antimicrobial agents [59,60,61,62,63,64,65,66,67,68,69,70,71,72,73,74,75,76,77,78,79,80,81,82,83,84,85,86,87]. Strategies and techniques have been developed for the exploitation of microalgae as expression systems for a number of advanced genetic toolkits [62,63,64,65].

Recombinant protein production was attained by engineering both nuclear and chloroplast microalgal genomes [66]. Chloroplast manipulation produces a high level of transgene expression [66,67,68,69,70,71], but the recombinant proteins obtained could be not always subjected to critical post-translation modifications (PTMs), thus affecting their activity and functionality [63]. On the other hand, nuclear transformation produced a lower protein accumulation, but retained PTMs, such as N-glycosylation [59,67]. Furthermore, nuclear manipulation provided tools for the targeting of proteins into specific subcellular compartments or secreting them into culture media [72] while regulatory elements (promoters and terminators) and selectable markers were tested to identify improved strategies for higher levels of recombinant protein [15,73].

Recently, the utilization of different engineering strategies, many biopharmaceutical compounds have been effectively produced in microalgae, and *Chlamydomonas reinhardtii* has been widely used for this type of applications.

A list of engineered strains is shown in Table 2. Antigens, antibiotics, and other bioactive molecules against animal pathogens have been produced. Examples include swine fever virus [74], white spot syndrome virus [64,72,73], lethal shrimp yellow head virus [77,78], *Aeromonas hydrophila* bacteria [79], and vibriosis disease [80]. 

In addition, recombinant proteins for therapy against human diseases and pathogens have been produced in microalgae. Dong et al. [60] obtained a *Ch. reinhardtii* strain able to produce the antimicrobial peptide Mytichitin-A. This peptide—not toxic for human cells—caused the inhibition of bacteria growth at the concentration of 60–80 ug/mL [57]. An interesting example of antimicrobial peptide against both fish and human pathogens is piscidin-4. This was recently produced in *H. pluvialis* by biolistic manipulation of chloroplast of this microalga [81].

Human interferon α (huIFN α) is widely used to treat cancer and some viral infections including hepatitis C; huIFN α was engineered and expressed in *Ch. reinhardtii* by *Agrobacterium* nuclear transformation [82]. The recombinant algae-expressed IFN α2 reduces virus propagation against the vesicular stomatitis virus [82].

Human interleukin-2 (iL-2), used in different cancers therapies, was successfully expressed in *Ch. reinhardtii* and *Dunaliella salina* producing about 0.94% and 0.59% of iL-2 respectively [68]. The stability of the recombinant algae-made interleukins was analyzed after five months from the transformation using ELISA assay, showing a stable and correct conformation of recombinant protein [68].

In recent years, microalgae were tested as suitable platforms for drugs production, even to counteract virus and other pathogens. Particularly, the receptor binding domain (RBD) of SARS-CoV-2, able to bind the angiotensin-converting 2 receptor (ACE2) to entry into human host cells was produced in *Chlorella vulgaris* and *Ch. reinhardtii* [83,84]. Thus, overexpressed RBD in algae was correctly folded, functional, and able to bind the human ACE2 receptor. These progresses open new pharmaceutical possibilities in the use of microalgae in the study and in the battle against the pandemic virus. Other aquatic microorganisms were similarly used as host organism in genetic engineering approaches for pharmaceutical purposes. Interesting results were obtained using the non-photosynthetic marine microorganism *Schizochytrium* sp. (Phylum: Heterokonta; Order: Labyrinthulales) [85,86,87].

Zika virus (ZIKV) infection cause severe symptoms including fever, rash, conjunctivitis, muscle and joint pains. ZK microalgae-made vaccine was tested by oral administration in mice showing immunogenic potential and inducing both systemic (IgG) and mucosal (IgA) humoral responses. Promising antigens against ZIKV were obtained by *Agrobacterium*-mediated transformation in *Schizochytrium* sp. [85].

RAGE (receptor of advanced glycation end-products) is a receptor related to development and progression of Alzheimer’s disease (AD), its expression was increased in the brain in AD patients, inducing the release of proinflammatory mediators; thus, RAGE is considered an interesting target for AD therapy [86]. RAGE fused to adjuvant carrier (LTB) was produced successfully in *Schizochytrium* sp. Furthermore, algae-based recombinant vaccine showed high stability to heat treatment (up to 60 °C), therefore, this was proposed as a possible vaccine for the treatment of AD [86].

CelTOS (cell-traversal protein for ookinetes and sporozoites) is crucial protein, which mediated the malaria virus infection. Shamriz et al. [65] manipulated *C. reinhardtii* chloroplast obtaining a CelTOS antigen from *Plasmodium falciparum* developing a sensitive ELISA test to identify malaria infection.

One of the great challenges is the development of an algae-based vaccines against cancer. Hernandez-Ramirez et al. [87] expressed in *Schizochytrium* sp. multi-epitope protein (BCB) against breast cancer (BC). They selected different B cell epitopes developed from tumor associated antigens (TAAs) recognized by the immune system. Biomass from transgenic algae expressing BCB were administered to mice, recording specific antibodies production [87]. 

Another goal is the production of monoclonal antibodies from microalgae. Nowadays, monoclonal antibodies represent an innovative solution against cancer and other human diseases. Vanier et al. [71] expressed human anti-hepatitis B (HBsAg) antibodies in the diatom *P. tricornutum*, reporting the production of fully assembled IgG antibody secreted in the culture medium. Microalgal antibodies were tested on THP-1 cell line to evaluate Fcγ receptor interactions, which are specific cell-superficial receptor for IgG, binding to human Fcγ receptor, and inducing an immune response. Monoclonal IgG antibodies against Marburg Virus (MARV), responsible for hemorrhagic fever with high fatality rates in Western Africa has been produced [88]. MARV sequences for both heavy and light chains from murine hybridoma were expressed in *P. tricornutum*, and the recombinant protein was secreted in the culture medium at concentration of 1300 ng/mL. ELISA demonstrated an efficient binding for algal antibodies [88].

In recent years, researchers explored the possible applications of extracellular vesicles (EVs) as drug carriers [89,90,91]. EVs are small cell-derived membranous particles delimited by a lipid-bilayer, containing proteins, lipids, and nucleic acids. EVs are released by cells into the environment under physiological and pathological conditions for cell communication [89,90]. EVs showed many interesting characteristics for potential use as therapeutic molecules and drug delivery systems. Indeed, EVs showed low immunological response, low toxicity, and these vesicles are able to cross various barriers, and showed the ability to transport and protect their cargo [89,92]. EVs have been evaluated for their potential to produce nanomedicine-based therapeutics; thus, the research has been focused on the possibility to use engineered EVs as vectors in specific therapies [91]. EVs from 18 different strains of microalgae have been isolated and characterized [92]. The EVs were approximately 90–160 nm in size, and microalgae-based EVs protein showed positive reactivity for the target protein of the EVs. Moreover, EVs did not show toxicity on either normal and cancer mammalian cell lines. These results suggest that EVs produced by microalgae could represent promising tools for therapeutic purposes.

It should be noted that all these studies require critical control points, in order to carefully regulate growth conditions and the activation of specific metabolic pathways. Therefore, the possibility to utilize extremophilic organisms as advanced bioreactors able to synthetize compounds upon particular conditions is a promising strategy for these aims.

**Table 2 plants-10-01686-t002:** List of pharmaceutical products obtained by engineered microalgae

Organism host	Product	Application	Transformation Method	Localization	Outcome	Expression Yields	References
*Ch. reinhardtii*	E2 protein	Swine fever virus vaccine	Biolistic	Chloroplast	Strong immunogenic response in mice	1.5–2% TSP	[74]
*Ch. reinhardtii*	VP28	White spot syndrome virus vaccine	Glass bead	Chloroplast	Shrimp survival up to 87%	ND	[75]
*D. salina*	VP28	White spot syndrome virus vaccine	Glass bead	Chloroplast	59% protection rate	78 mg/100 culture	[76]
*Ch. reinhardtii*	VP28			Chloroplast		Up to 10% TSP	[69]
*Ch. reinhardtii*	Antiviral dsRNA	Yellow head virus RNAi-based vaccine	Glass beads	Chloroplast	Reduced mortality	Up to 16 ng dsRNA/L culture	[77]
*Ch. reinhardtii*	dsRNA-YHV	Yellow head virus antiviral	Glass beads	Nucleus	22% Shrimp survival	45 ng/100-mL culture	[78]
*Chlorella* sp.	AMPs Scy-hepc	*A. hydrophila* bacteria oral antibiotics	Electroporation	Nucleus	*In vitro* inhibitory effects on *A. hydrophila*; in vivo *S. macrocephalus*	Up to 0.078%TSP	[79]
*H. pluvialis*	Piscidin-4 peptide	Antibacterial activity	Biolistic	Chloroplast	Stable expression	ND	[81]
*Nannochloropsis sp*	OmpK fragment gene	Vibrio species oral vaccine		Nucleus	Fifth generation stable immunogenic peptide production	ND	[80]
*Ch. reinhardtii*	Mytichitin-A peptide	Antibacterial activity	Electroporation	Nucleus	High inhibition of bacteria growth (MIC assays); No toxicity on HEK293 cells.	0.28% TSP	[60]
*Ch. reinhardtii*	SARS-CoV-2-RBD	Antigen proteins against SARS-CoV-2	Geminiviral vector	Transient	ELISA assay showed specific binding with the anti-RBD antibody	1.14 µg/g	[83]
*Ch. vulgaris*	SARS-CoV-2-RBD	Antigen proteins against SARS-CoV-2	Geminiviral vector	Transient	ELISA assay showed specific binding with the anti-RBD antibody	1.161 µg/g	[83]
*Ch. reinhardtii*	SARS-CoV-2-RBD	Antigen proteins against SARS-CoV-2	Electroporation	Transient	Bind human ACE2 receptor	0.1% TSP	[84]
*Ch. reinhardtii*	Human Interferon-α	Chronic viral diseases and cancers treat	*Agrobacterium*	Nucleus	In vivo e in vitro antitumoral activity, anticancer proprieties, antiviral activity	ND	[82]
*Ch. reinhardtii,*	Human interleukin-2	Interleukin production	*Agrobacterium*	Nucleus	ELISA assay showed biological activity, high stability	Up to 0.94% TSP	[72]
*D. salina; C. vulgaris*	Human interleukin-2	Interleukin production	*Agrobacterium*	Nucleus	ELISA assay showed biological activity, high stability	Up to 0.59% TSP	[72]
*Schizochytrium* sp.	ZK antigen	Zika virus oral vaccine	Algevir technology	Transient	IgG and IgA production	Up to 365.3 μg g^−1^ FW	[85]
*Schizochytrium* sp.	LTB:RAGE antigen	Alzheimer’s disease vaccine	Algevir technology	Transient	ELISA assay showed high stability up to of 60 °C	Up to 380 μg g^−1^ FW	[86]
*Ch. reinhardtii,*	PfCelTOS antigen	Malaria antigen for diagnosis tests	Biolistic	Chloroplast	Stable expression	ND	[64]
*Schizochytrium* sp.	Multiepitope protein (BCB)	Breast cancer vaccine	Algevir technology	Transient	Tumor cell line 4T1 reactivity; IgG production in mice immunized with BCB	Up to 637 μg/g	[87]
*P. tricornutum*	Hepatitis B Antibody	Antibodies against Hepatitis B	Biolistic	Nucleus	Binding FcγRI	2 mg/L	[71]
*P. tricornutum*	Monoclonal antibodies	Antibodies against Marburg virus	Biolistic	Nucleus	Elisa assay showed binding efficiency	1300 ng/ml	[88]
*T. pseudonana*	Antibody for EA1	Biosensor for anthracis detection	Biolistic	Nucleus	Detection of detected EA1 epitope in lysed spores	ND	[61]

## 4. Extremophilic Microalgae as Bioreactors

A renewed attention has been focused on organisms from extreme environments [93,94]. This is particularly true for microalgae, which show a wide range survival capabilities under extreme and stress conditions, namely hypersalinity, high and low temperatures or toxic heavy metals levels [94,95,96,97]. The use of extremophilic microalgae showed notable benefits for various applications [93,98]. Commercial cultivations of microalgae are usually obtained by growth in photobioreactors and open ponds [54,98]. On the other hand, industrial biotechnology requires elevated temperatures, difficult sterilization procedures, and expensive downstream processing which benefit from the use of thermophilic microalgae [99,100]. Extreme operating conditions tolerate by specific microalgae strains were proposed to improve these industrial processes, thus overcoming contaminations, loss of biomass, meteorological constraints, and others [94,101,102]. Example of biotechnological applications are the cultivation of *Chlorella* to mitigate the effects of industrial pollution, by absorbing CO_2_; the use of *Arthrospira platensis* for bioremediation of contaminated effluents; the utilization of a number of microalgal species for biofuel productions [100].

Thermophilic microalgae cultivated in large-scale open ponds are able to produce bulk amount of lipids, which can be utilised for biodiesel production technology [103]. Particularly, in recent years increasing attention were posed on the regulation of fatty acid dehydrogenase (FAD) in freeze resistant microalgae [59,104,105]. This studies contributed to both elucidation of the regulation of algal membrane assembly, and fatty acid metabolism. Promising results were obtained using the Δ12FAD from *Chlamydomonas* sp. ICE-L [59] and the Δ5FAD from *Lobosphaera incise* [105]. Both enzymes showed cold-resistant mechanisms to adapt their biochemical functions. Δ12FAD from *Chlamydomonas* sp. ICE-L contributes to membrane fluidity for adaptation to Antarctic extreme environment; Δ5FAD contributes to arachidonic acid synthesis, thus avoiding the risk of photodamage upon chilling. A number of snow algae were tested for their ability to produce lipid species in low temperatures [106]. These analysis reported an enriched production of unsaturated fatty acyl chains, especially C18:1n-9 and C18:3n-3, thus indicating these species as good candidates to improve the yields of microalgal biomass and oil products at low temperatures [106].

Thermophillic microalgae are studied for the applications of phycobiliproteins [107], a group of water-soluble proteins linked to chromophores, involved in light-harvesting processes. Phycobiliproteins find use in a wide range of commercial purposes (e.g., colorant for food and textile industries) and the exploitation of these proteins would give additional value [97]. A number of industrial processes require temperatures exceeding the stability of proteins from mesophiles [107]. Phycocianins from the cyanobacterium *A. platensis* are actually produced as dye in food industries but high temperatures (62 °C) limited their use [108]. Different authors proposed the use of phycocianins from the thermoacidophilic red alga *Cyanidioschyzon merolae* to overcome this problem [97,109]. The temperatures and pH tolerability of phycocianins from *Cyanidioschyzon merolae* were recently characterized by Yoshida et al. [96], confirming their thermotolerant properties. Similar evaluations were reported for phycobiliproteins in textile industry where the use of mesophiles proteins at high temperatures resulted in a decrease of color intensity [97].

A number of extromophilic microalgae were recently characterized for the ability in the production of bio-compounds with high economic interest [98]. For example, the snow green algae namely *Chlamydomonas nivalis*, *Raphidonema* sp., and *Chloromonas* sp. showed ability in the production of astaxanthin, α-Tocopherol, xanthophylles, and glycerol [110,111,112]. On the other hand, the heat tolerant green alga *Desmodesmus* and the acidophilic red alga *Galdieria sulphuraria* were tested for the production of lutein and blue pigment phycocianin, respectively [113,114]. These results demonstrate the unexplored potential represented by microalgae from extreme environments, thus encouraging the characterization of new species.

## 5. Conclusions

Here, we reviewed the remarkable value of photosynthetic organisms and their potential emerged in recent years. Through the last decades, there has been significant research into the exploitaion of microalgae as sustainable resources for the production of high-value products and proteins. It is clear that microalgae can play a crucial role in modern biotechnological processes role at energetic, pharmaceutical, and nutraceutical levels. It is worth to point out that even if the current knowledge about natural and engineering microalgae is broad and wide, present notions are still not sufficient, and significant efforts are required to reach the goals. The cloning technologies to obtain suitable platform or bio-factories to produce proteins or metabolites of interest are now widely accessible and at least readily implementable by modern technologies as well as synthetic biology approaches. It is now time to further expand this research and the applications of microalgae in order to develop more refined tools and strategies for an efficient sustainable and ecological breakthrough.

## Figures and Tables

**Figure 1 plants-10-01686-f001:**
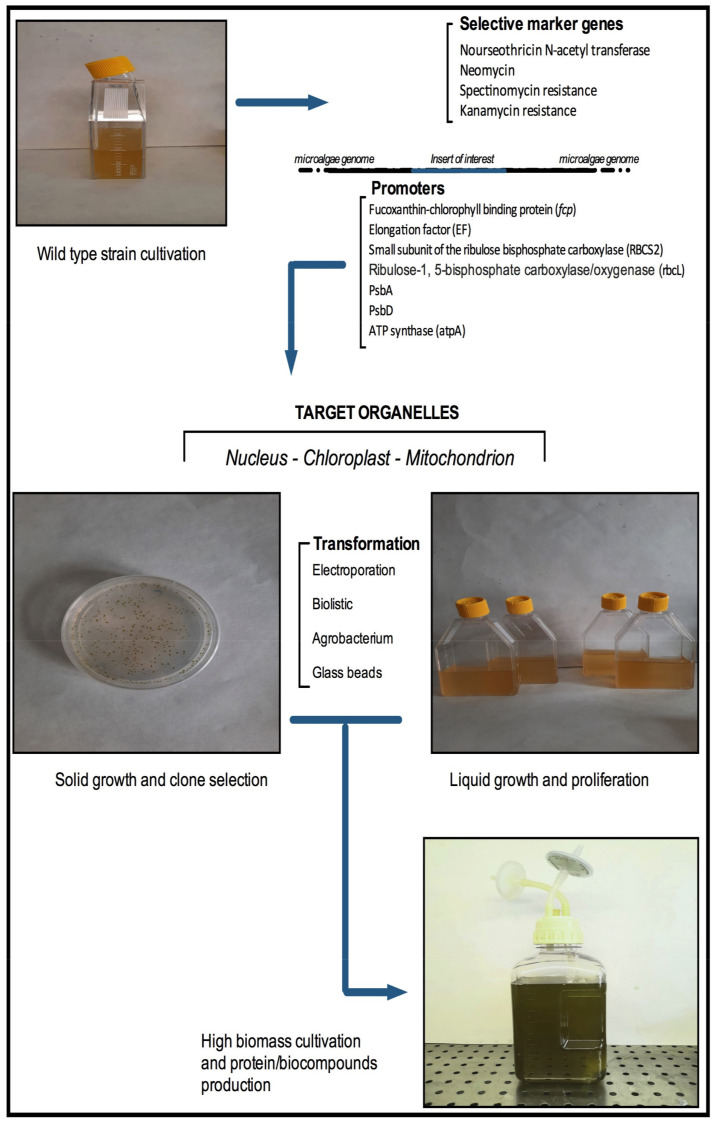
Recognized approaches (Transformation techniques, promoters, selective markers) used to microalgae genetic engineering.

## Data Availability

Not applicable.

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
