# Peer review of "Advanced Applications for Protein and Compounds from Microalgae"

_plants, 2021, doi:10.3390/plants10081686_

Round 1
Reviewer 1 Report
Dear authors,
the topic is very interesting and enormous large. Therfeore, many things can be added in this review, but i understand, that some limits have to be kept.
One serious remark is that diatoms are microalgae and there is no reason to write "macroalgae, microalgae and diatoms" (line 134, 136, Table 1). Diatoms are specific group of microalgae, but taxonomically their separation is not correct.
In table 1 and table 2: everywhere "sp." in Latin names should not be in Italic
Author Response
Dear authors, the topic is very interesting and enormous large. Therefore, many things can be added in this review, but i understand, that some limits have to be kept.
One serious remark is that diatoms are microalgae and there is no reason to write "macroalgae, microalgae and diatoms" (line 134, 136, Table 1). Diatoms are specific group of microalgae, but taxonomically their separation is not correct.
R1 response: We appreciate the suggestion of Reviewer #1. We corrected the manuscript as suggested, in the new version we avoided the distinction between “microalgae” and diatoms.
In table 1 and table 2: everywhere "sp." in Latin names should not be in Italic.
R1 response: Corrected
Reviewer 2 Report
This review by Castiglia and co-workers is a thorough work on the proteins derived from plants and algae, and their applications. I actually learned a lot reading this review, and thus I do not feel so qualified to judge it let's say from the science perspective. I am thus just going to comment on the text itself. I enjoyed the reading, although the authors should check spelling troughout the manuscript. Also, there are in each parts a lot of small paragraphs, I would suggest the authors to maybe reorganize the text into a limited number of paragraphs, for the moment it feels like all the ideas in each paragraphs are disconnected from each other. Finally some transitions between the different parts could be made, so that we can see the logical unfolding of the ideas presented in the text. Altogether this is I think a good review, but the authors could work a bit more on the text to make it more easy to read, and to make the story line more clear.
Author Response
R1 response: We thank the Reviewer#2 for these comments. In the new version of the manuscript, we reorganized text and English fluency to improve the story line. We preferred to keep the current division in paragraphs in order to help the Readers in selecting specific topic in the review. Connection sentences were added at the end of paragraphs to link the different topics of the review (blu highlight).
Reviewer 3 Report
This review analyses recent trends in the genetic technologies of a few microalgae, mainly in the food, and pharmaceutical levels. Important cultivated species are studied, namely Pheodactylum tricornutum, Chlorella vulgaris, Chlamydomonas reinhardtii, Dunalliela salina, among others. Enlightening tables are provided to summarize extensive information on these species.
It is an interesting review with good potential. A comprehensive study has been performed on the subject proposed by the authors, with interesting papers referred – many review papers, with broad applications of microalgae. The bibliography is recent and balanced.
Yet, within the paper, there are several mistakes that require to be addressed that are stated below.
The title is too broad and should refer specifically to microalgae only.
The abstract is a good representation of the review.
The text is interesting, updated, and relevant to the topic.
Yet, although it is always very difficult to address all the research, being a review paper on “microalgae” I believe that some other species should be addressed, being currently widely used in the microalgal industry. Nannochloropsis (fuel), Porphiridium (phycoerythrin), Haematococcus pluvialis (Astaxanthin), among others, have also been genetically transformed and should be addressed.
Thus, I support the publication of the paper proposed, with major reviewing, addressing further species.
Specific remarks are done below:
Title:
I advise the authors to change the title that is too broad, addressing “only” the microalgae sector.
Line 44 – diatoms are included in the informal designated microalgae group, being also invisible to the naked eye. The authors should clear to what taxonomic groups they are referring – otherwise, they can state “microalgae” in general. In fact, authors reference 9 states the taxa: Cyanobacteria (procaryotic) and different eukaryotic divisions (Chlorophyta, Rhodophyta, Pchrophyta including Bacillariophyceae (diatoms), and Chrysophyceae). But there are others (Dinoflagellata, Cryptophyta, Haptophyta, …).
Thus, I suggest the authors define microalga as “prokaryotic and eukaryotic photosynthetic microorganism” (reference 9) and just use this name.
Line 86 – correct spp to spp.
Line 90 – add to lettuce its scientific name (Lactuca sativa).
Line 94 – because tobacco is also a Nicotiana, the phrasing is deceiving. I suggest correcting the end of the sentence with: tobacco and other Nicotiana species.
Line 115 – already stated the scientific name of rice in the upper line, thus Oryza sativa can be deleted.
Line 132 – correct prpoposed
Line 134 – “microalgae” only, throughout the text instead of “microalgae and diatoms”
Line 140 – to what marine organism are the authors referring to? Actually, there are also terrestrial organisms, and other habitats, besides aquatic. Many microalgae (including diatoms) are also freshwater organisms; thus, the authors must specify the taxa they are referring to. Otherwise, the authors may refer to “aquatic organisms”.
Figure 1 – the photos are too stretched, please try to square them.
Agrobacterium should be in Italic.
Line 155 – if the authors are unaware of the Synechocystis species, they should state Synechocystis sp.
None of the references [61-63] cite this species, but reference [85] does. Thus, correct this.
Line 156 - Phaeodactylum and Thalassiosira are both diatoms – this is right.
Line 183 – correct enhnaced
Line 200 - This statement is not entirely true, for the concentration and lipid profile vary considerably among microalgae. The authors must be more specific – to which groups are they referring to regarding EPA and DHA?
Line 207 – references [78 Wang et al., 2019] and Hamilton et al. [2016] not according to the “Plants” template.
From this line on, references are not correct: e.g. the reference that states phytate as an anti-nutritional factor is 81. Pudney, A.; Gandini, C.; Economou, C.K.; Smith, R.; Goddard, P.; Napier, J.A.; Spicer, A.; Sayanova, O. Multifunctionalizing the marine diatom Phaeodactylum tricornutum for sustainable co-production of omega-3
Not:
- Nasopoulou, C. & Zabetakis, I. Benefits of fish oil replacement by plant originated oils in compounded fish feeds. A review. 568 LWTFood Sci Technol. 2012, 47, 217–224, https://doi.org/10.1016/j.lwt.2012.01.018.
Line 210 – correct “showwed”
Table 1 and 2. include the generic epithet the first time you write a new species, so the reader knows what the authors are referring to. The C. in C. vulgaris is Chlorella but in C. reinhardtii is Chlamydomonas.
Line 239 – Chlamydomonas reinhardtii in full name. The first time you refer to a species, use its full name. Then, you can use the small form C. reinhardtii. But because you have two different species with the same capital letter, you should use different letters. E.g. C. for Chlorella and Cy. or Ch. for Chlamydomonas.
Line 254 – Dunaliella salina?
Line 271 – correct incresed
Line 269 – Schyzochytrium and all the other quitrids are not, in fact, microalgae. They are non-photosynthetic microorganisms (although often stated as microalgae, and cultivated in heterotrophic devices as being microalgae).
Line 330 – the currently accepted name of Spirulina is Arthrospira. The authors should correct it.
Line 357 – write C. full name and in Italic.
Line 369 – microalgae, not algae
Line 372 – except the quitrids referred above …
Author Response
Reviewer #3
This review analyses recent trends in the genetic technologies of a few microalgae, mainly in the food, and pharmaceutical levels. Important cultivated species are studied, namely Pheodactylum tricornutum, Chlorella vulgaris, Chlamydomonas reinhardtii, Dunalliela salina, among others. Enlightening tables are provided to summarize extensive information on these species. It is an interesting review with good potential. A comprehensive study has been performed on the subject proposed by the authors, with interesting papers referred – many review papers, with broad applications of microalgae. The bibliography is recent and balanced.
R1 response: We thank the Reviewer#3 for these comments. As requested, we prepared a new version of the manuscript. Corrections are in blue.
Yet, within the paper, there are several mistakes that require to be addressed that are stated below.
The title is too broad and should refer specifically to microalgae only.
R1 response: We corrected the title as suggested by Reviewer #3. The new title is “Advanced Applications for Protein and Compounds from microalgae and plants”.
The abstract is a good representation of the review.
The text is interesting, updated, and relevant to the topic.
Yet, although it is always very difficult to address all the research, being a review paper on “microalgae” I believe that some other species should be addressed, being currently widely used in the microalgal industry. Nannochloropsis (fuel), Porphiridium (phycoerythrin), Haematococcus pluvialis (Astaxanthin), among others, have also been genetically transformed and should be addressed.
R1 response: As requested by Reviewer #3, we investigated about the recent publication on the indicated species.
Thus, I support the publication of the paper proposed, with major reviewing, addressing further species.
Specific remarks are done below:
Title:
I advise the authors to change the title that is too broad, addressing “only” the microalgae sector.
R1 response: We corrected the title as indicated by reviewer #3. The new title is “Advanced Applications for Protein and Compounds from microalgae and Plants”.
Line 44 – diatoms are included in the informal designated microalgae group, being also invisible to the naked eye. The authors should clear to what taxonomic groups they are referring – otherwise, they can state “microalgae” in general. In fact, authors reference 9 states the taxa: Cyanobacteria (procaryotic) and different eukaryotic divisions (Chlorophyta, Rhodophyta, Pchrophyta including Bacillariophyceae (diatoms), and Chrysophyceae). But there are others (Dinoflagellata, Cryptophyta, Haptophyta, …).
R1 response: In the new version of the manuscript we avoided the distinction between “microalgae” and diatoms.
Thus, I suggest the authors define microalga as “prokaryotic and eukaryotic photosynthetic microorganism” (reference 9) and just use this name.
R1 response: In the new manuscript version this definition is in the introduction to define the entire microalgae group.
Line 86 – correct spp to spp.
R1 response: Done
Line 90 – add to lettuce its scientific name (Lactuca sativa).
R1 response: Done
Line 94 – because tobacco is also a Nicotiana, the phrasing is deceiving. I suggest correcting the end of the sentence with: tobacco and other Nicotiana species.
R1 response: Corrected
Line 115 – already stated the scientific name of rice in the upper line, thus Oryza sativa can be deleted.
R1 response: Corrected
Line 132 – correct prpoposed
R1 response: Corrected
Line 134 – “microalgae” only, throughout the text instead of “microalgae and diatoms”
R1 response: Corrected
Line 140 – to what marine organism are the authors referring to? Actually, there are also terrestrial organisms, and other habitats, besides aquatic. Many microalgae (including diatoms) are also freshwater organisms; thus, the authors must specify the taxa they are referring to. Otherwise, the authors may refer to “aquatic organisms”.
R1 response: We replaced marine organism with aquatic organism
Figure 1 – the photos are too stretched, please try to square them.
R1 response: A new improved version of Figure 1 is the in R1 version of the manuscript.
Agrobacterium should be in Italic.
R1 response: Corrected
Line 155 – if the authors are unaware of the Synechocystis species, they should state Synechocystis sp.
R1 response: Corrected
None of the references [61-63] cite this species, but reference [85] does. Thus, correct this.
R1 response: Thank you. We corrected citing reference 62.
Line 156 - Phaeodactylum and Thalassiosira are both diatoms – this is right.
Line 183 – correct enhanced
R1 response: Corrected
Line 200 - This statement is not entirely true, for the concentration and lipid profile vary considerably among microalgae. The authors must be more specific – to which groups are they referring to regarding EPA and DHA?
R1 response: We cited some microalgae species used for PUFA production.
Line 207 – references [78 Wang et al., 2019] and Hamilton et al. [2016] not according to the “Plants” template.
R1 response: Corrected
From this line on, references are not correct: e.g. the reference that states phytate as an anti-nutritional factor is 81. Pudney, A.; Gandini, C.; Economou, C.K.; Smith, R.; Goddard, P.; Napier, J.A.; Spicer, A.; Sayanova, O. Multifunctionalizing the marine diatom Phaeodactylum tricornutum for sustainable co-production of omega-3
Not:
Nasopoulou, C. & Zabetakis, I. Benefits of fish oil replacement by plant originated oils in compounded fish feeds. A review. 568 LWTFood Sci Technol. 2012, 47, 217–224, https://doi.org/10.1016/j.lwt.2012.01.018.
R1 response: Reference [former 80, now 83] is Nasopoulou, C. and Zabetakis while reference [former 81 now 84] is Pudney et al. In order to avoid misunderstanding the sentence has been rephrased.
Line 210 – correct “showwed”
R1 response: Corrected
Table 1 and 2. include the generic epithet the first time you write a new species, so the reader knows what the authors are referring to. The C. in C. vulgaris is Chlorella but in C. reinhardtii is Chlamydomonas.
R1 response: Now full name of species were in table 1.
Line 239 – Chlamydomonas reinhardtii in full name. The first time you refer to a species, use its full name. Then, you can use the small form C. reinhardtii. But because you have two different species with the same capital letter, you should use different letters. E.g. C. for Chlorella and Cy. or Ch. for Chlamydomonas.
R1 response: Corrected
Line 254 – Dunaliella salina?
R1 response: Corrected
Line 271 – correct incresed
R1 response: Corrected
Line 269 – Schyzochytrium and all the other quitrids are not, in fact, microalgae. They are non-photosynthetic microorganisms (although often stated as microalgae, and cultivated in heterotrophic devices as being microalgae).
R1 response: In the new version of the manuscript, we specify that Schyzochytrium sp. are not conventional microalgae to maintain the notions about these interesting marine species.
Line 330 – the currently accepted name of Spirulina is Arthrospira. The authors should correct it.
R1 response: Corrected
Line 357 – write C. full name and in Italic.
R1 response: Corrected
Line 369 – microalgae, not algae
R1 response: Corrected
Line 372 – except the quitrids referred above …
R1 response: Corrected
Round 2
Reviewer 3 Report
I acknowledge the extensive changes made by the authors to improve the paper. The authors have thoroughly reviewed the paper, making all the changes suggested, thus the paper has been much improved.
New species were added to the paper, making this review broader and thus more interesting to read.
Thus, I recommend the publishing of the paper in the present form.
Author Response
Thank to reviewer #3 for the suggestions.